# Adsorption and Desorption Behaviour of Polychlorinated Biphenyls onto Microplastics’ Surfaces in Water/Sediment Systems

**DOI:** 10.3390/toxics8030059

**Published:** 2020-08-17

**Authors:** Marta Llorca, Manuela Ábalos, Albert Vega-Herrera, Miquel A. Adrados, Esteban Abad, Marinella Farré

**Affiliations:** Institute of Environmental Assessment and Water Research (IDAEA-CSIC), C/Jordi Girona, 18–26, 08034 Barcelona, Spain; mlcqam@cid.csic.es (M.L.); manuela.abalos@idaea.csic.es (M.Á.); avhqam@cid.csic.es (A.V.-H.); miquel.adrados@idaea.csic.es (M.A.A.); eaheco@idaea.csic.es (E.A.)

**Keywords:** microplastics (MPLs), polyethylene, polystyrene, polyethylene terephthalate, polychlorinated biphenyls, adsorption, marine microcosm

## Abstract

The potential of microplastics (MPLs) in marine ecosystems to adsorb and transport other micropollutants to biota, contributing to their entry in the food chain, is a primary cause of concern. However, these interactions remain poorly understood. Here, we have evaluated the adsorption/desorption behaviour of marker polychlorinated biphenyls (PCBs), onto MPL surfaces of three widely used polymers—polystyrene (PS), polyethylene (PE), and polyethylene terephthalate (PET). The range of MPL sizes ranged from 1 to 600 μm. The adsorption/desorption was evaluated in sediment/water systems in marine microcosms emulating realistic environmental conditions for 21 days. The adsorption percentages ranged from 20 to 60%. PCBs with a lower degree of chlorination showed higher adsorption percentages because of conformational impediments of PCBs with high-degree chlorination, and also by their affinity to be adsorbed in sediments. Glassy plastic polymers as PET and PS showed a superior affinity for PCBs than rubbery polymers, such as PE. The polymers that can bond PCBs by π-π interactions, rather than van der Waals forces showed better adsorption percentages, as expected. Finally, the adsorption/desorption behaviour of selected PCBs onto MPLs was fitted to a Freundlich isotherm model, with correlations higher than 0.8 in most of the cases.

## 1. Introduction

Some of the primary problems associated with the presence of plastic litter in marine environments are their behaviour as a source and transfer vectors for co-contaminants to the aquatic food web. These impacts are partially influenced by their size. Microplastics (MPLs), defined as plastic pieces below 5 mm, including nanoplastics (NPLs), is a different environmental problem compared to macro- and meso-plastic pollution. MPLs can enter in the environment in the small-size range (classified as primary MPLs) or can be generated once in the environment by fragmentation and erosion of plastic pieces and debris (classified as secondary MPLs). MPLs/NPLs, due to their small size, similar to plankton, can be ingested by aquatic organisms, and therefore be introduced into the marine food web [1,2,3]. It should be highlighted that fish, bivalves, or mammals cannot digest MPLs since they do not have enzymatic pathways available to break down the synthetic polymers. However, these particles can be retained in some organs [4], and the nanoparticles, due to their small size, can be translocated in living tissues with adverse effects. It is estimated that some of the plastics can reach concentration factors inside the organisms near to a 1 million-fold increase [5]. The interaction between polymers and contaminants is a very complex problem that remains poorly understood. Therefore, realistic risk assessment studies to characterize and evaluate this type of interaction are highly required to establish measures to minimize the negative effects of MPLs in coastal environments. The adsorption/desorption behavior among MPLs and co-contaminants is influenced by different factors, such as the type of co-contaminant, the polymeric matrices of MPLs (e.g., polyethylene, polystyrene, or a combination of different monomers), their relative concentrations, and the environmental conditions. However, some trends have been identified, such as the high capacity to adsorb and accumulate hydrophobic organic chemicals (HOCs) on their surface from the surrounding areas [6,7,8,9]. In consequence, and in addition to the already defined routes of exposure to HOCs, MPLs could act as a back door to the entrance of these contaminants to organisms once they are adsorbed in the polymer surface [2,6,10,11,12] and the human food chain. 

Polychlorinated biphenyls (PCBs) are a group of manmade HOCs that were widely used in the past in electrical equipment. They are currently forbidden and, since 2001, the 209 congeners are listed in the Stockholm Convention of persistent organic pollutants (POPs) [13]. Although they have been banned, PCBs are still present in the environment and are present ubiquitously in biota [14] and sediments [15]. 

The adsorption of PCBs in MPLs from natural environments has been informed [16]. The uptake and incorporation of these contaminants from MPLs to biota have also been assessed [17]. For example, it has been observed that PCBs adsorbed on polystyrene (PS) and polyethylene (PE) were significantly bioaccumulated in Norway lobster [11], while the bio-uptake of sediment invertebrate worms from contaminated polypropylene (PP) with PCBs was low, but not negligible [10]. 

On the other hand, some studies conducted under controlled conditions assessed the adsorption of organic contaminants onto MPL surfaces [8,9]. In the case of PCBs, the parameters for the adsorption of PCBs have been studied to micro-PE and nano-PS in a system composed of plastic particles, PCBs, and water [18,19]. However, complementary studies in more realistic environmental scenarios, including sediments competing with the MPLs and the suspended organic material, are necessary to describe the adsorption/desorption behaviour of HOCs on MPL surfaces in aquatic systems. 

In this regard, the main objective of this work was to characterize the adsorption/desorption capacity of MPLs of polystyrene (PS), high-density polyethylene (HDPE), and polyethylene terephthalate (PET) (sizes between 1 and 300 μm) for seven commonly detected polychlorinated biphenyls (PCBs: 28, 52, 101, 138, 153, 180, and the dioxin-like PCB 118), that are used as contamination indicators. The experiments were designed for a water/sediment system at environmental conditions and relevant concentrations of PCBs. Since amorphous and semi-amorphous plastics are less resistant to chemical attack and environmental stress [20], PS was chosen as an example of amorphous thermoplastic. On the other side, PE and PET were selected because of crystalline or semi-crystalline structures that make them resistant to chemical attack [20]. Furthermore, the diffusion capacity of glassy plastic polymers (i.e., PET and PS), which used to have less diffusion than non-glassy or rubbery polymers (i.e., PE) [21], will be evaluated.

## 2. Materials and Methods 

### 2.1. Materials

MPLs of HDPE microspheres ranging from 3 to 16 µm were supplied by Cospheric (Santa Barbara, CA, USA); 10 µm PS, supplied by Phosphorex (Hopkinton, MA, USA); and PET microspheres below 300 µm from GoodFellow (Huntingdon, UK). 

Mixtures of seven native Marker PCBs (IUPAC nos. 28, 52, 101, 118, 138, 153, and 180) (BP-D7) and labelled PCBs (MBP-MXE; P48RS), as well as P48-M-CV calibration solutions for Marker PCBs were supplied by Wellington Labs (Guelph, ON, Canada). These standards were used for spiking experiments (spiking mix prepared in methanol at 10 µg/mL), quantification, analytical recovery calculation, and calibration.

Solvents used during sample treatment and analysis were of high purity and supplied from J. T. Baker (JT Baker, Phillipsburg, NJ, USA): methanol, acetonitrile, dichloromethane, n-hexane, ethyl acetate, n-nonane, and water. Sodium chloride ≥ 99% was purchased from Merck (Madrid, Spain).

### 2.2. Experimental Design 

Mixtures of seawater and real sediments that were previously analysed, characterised, and fortified at different concentrations with the PCB mixtures were used to carry out the adsorption experiments under environmentally relevant conditions. 

Sediment samples of Alfacs Bay (Ebro Delta, Catalonia, NE of Spain; 40.622999, 0.661707) were obtained. The samples were collected in aluminium foil trays and transported to the laboratory in cold conditions. Once in the lab, 500 g of sediment was dried at room temperature (25 °C) under a fume cupboard, then sieved and homogenised. Afterwards, the background concentration of marked PCBs was characterised according to Section 2.3 and Section 2.4. 

Hereafter, six sediment subsamples of 4.5 g were separated, and they were spiked to the following concentrations of the PCB mixture (0, 5, 10, 15, and 25 ng/g d.w spiked to the appropriate volume of a mixture of 10 µg/mL in methanol). The subsamples were left to reach the equilibrium in a desiccator for 24 h. After that time, the sediments were homogenised again, and three series of six tubes, one tube with each PCB concentration per set was prepared. In each tube, 4.5 g of sediment was introduced. Then, 2 µg/g of each type of MPL (PE, PS, and PET) was added to each tube. In addition, two more extra tubes were prepared, one tube with sediment without a PCB spike and MPLs, and another tube with sediment and PCBs (spiked at 25 ng/g d.w) but without MPLs. A schematic flowchart can be seen in Appendix A.

All the tubes were shaken vigorously in a vortex for 5 min and maintained at room temperature in an orbital digester at 40 rpm for 2 days. Finally, 10 mL of pristine seawater, previously analysed and characterised, was added to each tube. These mixtures were then homogenised for 1 min in a vortex. These seawater/sediment systems were sealed and agitated at room temperature for 21 days in an orbital digester at 40 rpm under dark conditions to avoid the photo-degradation of the organic matter.

After this time, the seawater/sediment systems were saturated with NaCl in order to increase the water density. Floatable MPLs were separated manually with a nano-filter mesh from the upper layer of water, while the rest of the supernatant was separated by decantation. Then, the sediments were dried at room temperature and homogenised before PCB extraction.

### 2.3. PCB Extraction

About 0.5 g of each dry homogenised sediment were spiked with the labelled surrogate internal standard mixture (MBP-MXE). Then, PCBs were recovered from the sample by Soxhlet extraction for 24 h, with a mixture of n-hexane: dichloromethane (1:1, *v/v*). After that, the extract was concentrated in a rotary evaporator and transferred to n-hexane prior to the clean-up step. Purification and fractionation were carried out in open columns of multilayer silica (2:1; acid/base) and activated Florisil^®^ (at 600 °C) eluted with n-hexane. The final extract was concentrated again to ca. 1 mL in a rotary evaporator, transferred to a 2 mL vial, and evaporated under a nitrogen current. Finally, a known amount of the labelled recovery internal standard mixture (P48-RS) dissolved in n-nonane was added as an internal standard before instrumental analysis. The samples were analyzed by triplicates.

### 2.4. Analysis by Gas Chromatography Coupled to High-Resolution Mass Spectrometry (GC-HRMS)

The analysis was done in an Agilent Technologies 6890N gas chromatograph (Agilent, Palo Alto, CA, USA) coupled to a Micromass AutoSpec—Ultima NT (Waters, Manchester, UK) high-resolution mass spectrometer (EBE geometry) controlled by a Masslynx data system.

The chromatographic system was equipped with a DB-XLB (Agilent, Folsom, CA, USA) fused-capillary column (60 m × 0.25 mm I.D. × 0.25 μm film thickness). Following the temperature program that can be seen elsewhere [14]. The injection of 1 µL of extract was carried out in splitless mode (60 s) with the temperature of the injector at 280 °C. 

The chromatograph was coupled to a magnetic sector spectrometer equipped with an electron impact ionisation (EI) source working at 32 eV, trap current at 500 µA, and acceleration voltage at 8000 V. The acquisition was performed in selected ion monitoring (SIM) mode at a resolving power of 10,000 (5% valley). The ion source and transfer line were set at 250 and 280 °C, respectively.

The quantification of residual PCBs in sediments was done by isotopic dilution methodology described elsewhere [14].

## 3. Results and Discussion

### 3.1. Adsorption of Marker PCBs on Selected MPLs

In Table 1, the percentages of adsorption of the marker PCBs on selected MPL surfaces calculated according to Equation (1) are summarized:(1)$$\%\;\mathrm{Ads}=100-\frac{{\left[A\right]}_{t}}{{\left[A\right]}_{0}}\times 100$$
where % Ads is the percentage of adsorption, [A]*_t_* is concentration of compound A remaining in sediments after 21 days, and [A]_0_ is the concentration of compound A at time 0, both concentrations expressed in ng/g. The distribution of PCBs is schematized in Appendix A.

#### 3.1.1. Adsorption onto PS-MPLs 

In Table 1, the adsorption percentages for PCBs onto the PS surface is shown. The adsorption range was from 10 to 60% and ordered as follows: PCB-52 ≥ PCB-28 > PCB-101 > PCB-153 ≥ PCB-138 > PCB-180. As it can be seen, the congeners with more chlorine atoms in their molecules generally showed less adsorption to PS-MPL surfaces, with the exception of the dioxin-like compound, PCB 118 (dlPCB-118), possibly due to the (co-)planarity of its conformation (see Table 2) [22]. This specific conformation of dlPCB-118 could make difficult its adsorption onto the PS surface. In the case of structures with more chlorines, the lower adsorption compared with the other PCBs with fewer chlorines is due to the surface of the spatial structure of molecules with a high number of chlorines, which make difficult their stabilisation on plastic pore sites. This phenomenon was also observed by Pascall et al. [21] using PS film, where the diffusion coefficient decreased with the increase of molar volume of PCBs. The main hypothesis is that PCBs diffuse into PS particles through the static boundary layer around the PS particle [18]. This is an easier process for molecules with lower molar volume, like PCB-28 or PCB-52. This is a relatively slow process. First, PCBs are adsorbed on the particle surface by weak van der Waals interactions. Then, PCB molecules further diffuse inside the glassy polymeric structure of MPLs like PS [18,23]. Once there, their desorption is less favoured because they are physically entrapped [23]. In addition, the adsorption of highly chlorinated PCBs onto PS-MPLs might have been less favored because of the MPL-sediment competition. It has been reported that highly chlorinated PCBs have a high affinity with sediments. 

#### 3.1.2. Adsorption onto PE-MPLs 

As can be seen in Table 1, the percentage of adsorption of PCB-28 and PCB-52 onto PE was nearly 60% after 21 days. Both compounds, PCB-28 and PCB-52, are those with a lower chlorination degree of three and four chlorine atoms, respectively. It has been shown that the adsorption decreases with the increase of the number of chlorine atoms, the final adsorption being ranked as follows: PCB-28 ≈ PCB-52 > PCB-101 > PCB-153 > PCB-138 > PCB-180. Similar adsorption behaviour is also shown for PS. The differences observed among different congeners can be attributed to different effects at the same time. First, there is the steric interaction of PCB molecules with PE surfaces. PE is a nonporous material, with only transient cavities with a typical size of 1 nm [26], which is a limitation for the adsorption of large molecules or big cluster molecules with suspended materials that are excluded by size (Table 1). The aggregates formation between the suspended matter and HOCs as PCBs is expected to happen, especially for the more non-polar compounds [26]. The second factor influencing the lower adsorption onto PE of more chlorinated congeners is their much higher interaction with sediments, as has been reported. In summary, the adsorption onto PE surfaces of PCBs 118, 138, and 180, with high steric impediments due to chlorine number and position was limited, and at the same time, their interaction to sediments was higher. For example, for the PCB 118, the maximum adsorption level was 15% for the spiking of 15 ng/g, and the percentage of adsorption was even lower for higher concentrations of spike because of the high non-polarity of this compound that tends to form a cluster of molecules in seawater (Table 1). These results agreed with previous works, such as when using PE food-packaging plastic film [21], in which it was confirmed that there was a correlation between increasing chlorination and increasing cohesive density within PCB molecules [21]. This phenomenon made difficult the adsorption of molecules with a high number of chlorines. In another work, Allen et al. [27] evaluated the adsorption of PCBs 52, 101, 118, and 153, among others, in six plastic polymers, including high- and low-density PE, PS, and PET, where it was observed that PCBs from spiked waters were easily adsorbed onto PE and PS surfaces with a return of compounds after desorption with n-hexane between 50 and 55% [27].

#### 3.1.3. Adsorption onto PET-MPLs 

The results of adsorption of PCBs on a PET surface exhibited a similar profile than that observed for PS and PE (see Table 1). The adsorption capacity for selected PCBs decreased with the increase of chlorines in the molecule. The maximum adsorption is exhibited by PCB-28 (c.a. 70%), followed by PCB-52 (c.a. 60%). However, as observed for the other MPLs, the dlPCB-118 has the lowest tendency to be adsorbed on PET surface, its maximum being at 30% for 10 ng/g of spiking level and the minimum at 10% for higher spiking concentrations. 

#### 3.1.4. Comparison among MPLs 

Comparing adsorption percentages among MPLs, PET has higher percentages (generally being 10% higher than the others) for all spiking concentrations, as shown in Table 1. 

In this sense, the effects observed by Pascall et al. [21], where glassy polymers like PVC have less diffusion and lower adsorption capacity compared to non-glassy or rubbery polymers such as PE, is not confirmed here. In this work, PS and PET (glassy polymers) exhibited equal or even higher adsorption than PE, as it was also observed in a long-term exposition experiment by Rochman et al. [28]. The difference between our work and the one carried out by Pascall et al. [21] can be explained by experimental differences, such as the size of MPLs (1–300 µm in this work) and the type of material used. In the present study, we used granulated MPLs, while Pascall et al. [21] used films. 

The adsorption behaviour of PCBs onto the different MPLs materials was PET > PS > PE (Table 1). These results can be explained by the type of interaction between surfaces and PCBs. With the aliphatic chains of PE, only non-specific van der Waals interactions can be established, while PS and PET can also undergo π-π interactions, as pointed out by other authors [27,28].

### 3.2. Adsorption Isotherms of Marker PCBs on Selected MPLs

Different adsorption isotherms were tested, including the Brunauer–Emmett–Teller (BET), Langmuir, and Freundlich models [29]. The Freundlich model was shown to be the one which fit best, and it was selected in all the cases (Equation (2)). This model assumes the presence of an infinite number of different adsorption sites [30]. Then, the equation was linearized in its logarithmic form to obtain the Freundlich constants according to Equation (3):(2)$$q=K_{F}C^{n}$$
(3)$$logq=logK_{F}+\frac{1}{n}logc$$
where *q* is the concentration of PCBs onto the plastic surface and *C* the concentration of PCBs in sediment at the assumed equilibrium time of 21 days. *K_F_* is an indicator of the adsorption capacity, where the higher the value is, the higher the adsorption capacity. Additionally, *n* is a value ranging from 0 to 10, where an *n* range between 2 and 10 indicates good capacity of adsorption, between 1 and 2 means moderate adsorption capacity, and less than 1 indicates poor adsorption capacity at high concentrations of compounds [29]. In Figure 1 and Appendix A, the results are presented. 

In the case of PS-MPLs, as it can be seen in Table 3, their characteristic parameters indicate high adsorption capacity to PCBs with a high *K_F_* value, while the adsorption is favoured at the lower concentrations because n is inferior to 1. This agrees with the observations by Hüffer et al. [31] in their investigations of the adsorption of non-polar organic compounds onto MPLs. In that study, the non-linear adsorption isotherm in PS was attributed to surface adsorption. 

The adsorption capacity of PE and PET MPLs for the selected PCBs in water/sediment systems was shown to be similar to PS-MPLs. Additionally, as can be seen in the Figure 1B,C, and Appendix A, the Freundlich isotherm was the best-fitting equation for the interpretation of the results. In both cases, the adsorption is favoured at low concentrations, with *n* values inferior to 1 for PE-MPLs and similar to 1 for PET-MPLs. 

As can be seen for the three types of MPL materials, the congener 118 cannot be linearized, where in this case, their co-planarity (all chlorines are in the same plane) probably produced surface interaction, rather than introduction in pores or cavities. For the rest of the compounds, the steric congestion (presence of surrounding ligands (chlorines in PCBs) in a molecule) increase with the number of ortho-chlorines, which means that di-ortho compounds (52, 101, 153, 138, and 180) present a more steric congestion than the mono-ortho congener 28-PCB. Additionally, among the di-ortho compounds, the steric congestion increases with the total number of chlorines in the molecule. Proportional to the increase in steric congestion, the interaction will be more superficial than in the cavities, and then the adsorption isotherms will be less linear. Besides, when the number of chlorines increases, the non-polarity rises too, and the interactions between the PCBs and the sediments is stronger, and therefore, a lesser amount of molecules will be available to be adsorbed onto the MPLs. In spite of how the Freundlich model cannot explain the 100% of the interactions, in some particular cases, the model fits better to the globally studied system.

### 3.3. Environmental Implications 

The main results reported in this work suggest that recalcitrant compounds, such as PCBs, can be readily adsorbed onto cavities of PE-MPL surfaces, rather than inside the polymeric structure; while in PS-MPL and PET-MPLs, a certain degree of diffusion inside the polymeric structure occurred. In the case of PS and PET-MPLs, the π-π interactions also increased the adsorption rate of PCBs in comparison to the adsorption of PCBs onto PE-MPLs, in which the interactions were just van der Waals forces. We want to remark that in this work, we have tested the ability of three types of MPLs to adsorb PCBs in a water/sediment system, and despite the competition with sediments, the MPLs presented an important rate of accumulation. Due to the MPLs’ size and variety of colour that has been proved to stimulate their ingestion by fish, this can be a new route of HOC exposure to biota, and an entrance to the aquatic, and therefore to the human food chain. Furthermore, it is essential to study the fugacity gradient of MPLs with adsorbed PCBs between the water/sediment/aquatic food chain, since these are the main driving forces of transfer. Finally, although this study has been focused on seven marker PCBs, similar results are expected for other equally recalcitrant congeners.

## 4. Conclusions

Within this work, we have reported data regarding the adsorption capacity of MPLs for marker PCBs in water/sediment systems. To the authors’ knowledge, this is the first time that the adsorption capacity of MPLs for PCBs has been evaluated in this type of system.

The main results have shown that, after three weeks of exposition, the recalcitrant PCBs are adsorbed, in general, between 20% and 60% onto MPLs of PE, PS, and PET. The tested polymers showed that the adsorption is favoured when π-π interactions can be done like in the case of PET and PS. Finally, the calculated isotherms were fitted to a Freundlich equation with a Pearson coefficient higher than 0.9 for the sum of PCBs and, in almost all the cases, higher than 0.8 for individual PCBs. With these data, it can be predicted that PCBs dissolved at low ppt and sub-ppt concentrations will be readily and strongly attached to MPL surfaces, given that PCBs associated with plastic particles are likely to be a significant factor in the environmental fate, behaviour, and potential transfer to the food chain.

## Figures and Tables

**Figure 1 toxics-08-00059-f001:**
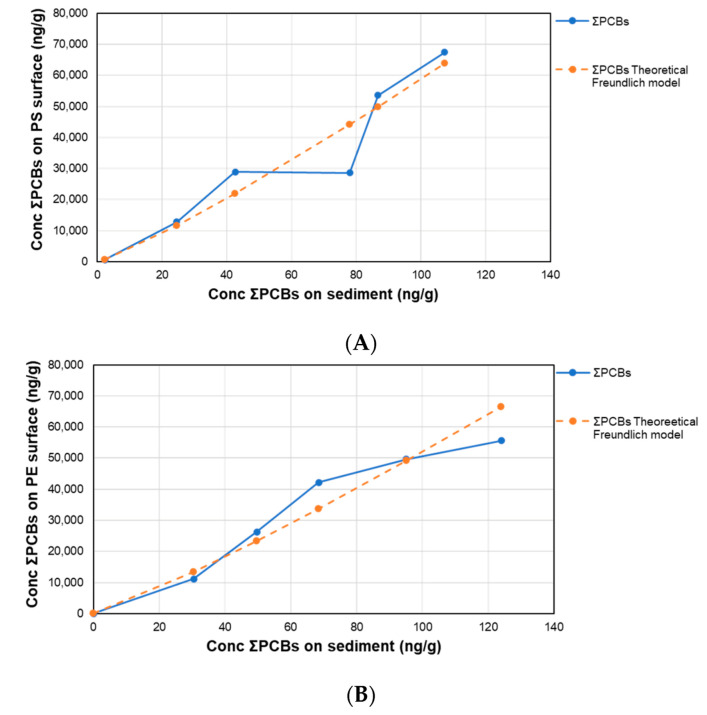
Adsorption isotherms of concentration (Conc) of ƩPCBs after 21 days on MPL surface of (**A**) PS, (**B**) PE, and (**C**) PET vs. sediment.

**Table 1 toxics-08-00059-t001:** Adsorption percentages for PCBs vs. spiked concentration in all polymers and sum of PCBs adsorbed on MPLs.

Spiking Level	PCB Congener	5 ng/g		10 ng/g		15 ng/g		20 ng/g		25 ng/g	
		Mean	±SD	Mean	±SD	Mean	±SD	Mean	±SD	Mean	±SD
%Ads on PE	PCB-28	58.27	5.48	58.24	4.08	60.64	2.43	58.59	1.76	60.88	6.70
	PCB-52	56.29	4.50	59.95	1.68	60.55	3.63	57.14	4.00	58.64	3.05
	PCB-101	41.28	2.06	42.72	2.01	44.93	0.81	41.22	3.67	38.89	1.56
	PCB-118	0.00	0.00	12.89	0.80	14.17	0.69	5.75	1.04	0.00	0.00
	PCB-153	29.03	2.03	36.48	1.09	38.27	2.68	34.21	1.54	29.64	2.67
	PCB-138	5.77	0.36	28.31	2.26	31.01	3.10	27.37	0.55	22.78	0.91
	PCB-180	16.22	0.29	14.21	1.28	28.13	1.60	24.82	2.63	18.61	1.38
ΣPCBs on PE (ng)		0.103		0.253		0.417		0.498		0.574	
%Ads on PET	PCB-28	62.87	4.21	66.96	2.68	63.82	2.55	63.53	10.80	65.47	1.64
	PCB-52	60.02	2.82	62.97	4.22	58.71	4.70	58.06	3.08	60.11	4.03
	PCB-101	46.46	0.74	53.15	3.93	49.49	3.32	47.43	3.32	46.16	5.08
	PCB-118	17.34	0.35	32.13	2.57	20.45	0.39	16.94	1.52	15.66	1.47
	PCB-153	32.81	1.64	39.10	1.96	35.26	1.69	32.93	0.49	31.94	1.53
	PCB-138	21.78	1.96	35.02	4.20	29.28	4.10	26.25	1.05	22.45	1.35
	PCB-180	21.87	1.77	26.09	1.54	26.71	0.99	25.23	4.04	20.98	1.47
ΣPCBs on PET (ng)		0.132		0.315		0.426		0.541		0.657	
%Ads on PS	PCB-28	48.58	0.97	56.12	6.17	50.13	6.52	54.86	4.39	53.46	4.81
	PCB-52	55.19	3.86	60.49	7.86	53.60	7.50	59.41	3.80	57.07	2.17
	PCB-101	38.16	3.05	45.74	3.66	32.49	2.08	41.74	1.17	40.97	4.26
	PCB-118	14.42	1.50	31.22	2.22	8.17	0.66	21.17	1.48	20.19	0.36
	PCB-153	33.45	0.47	41.80	4.51	25.82	1.50	37.81	4.54	38.74	2.60
	PCB-138	26.42	1.59	34.24	1.57	15.50	1.86	31.00	4.34	36.63	0.51
	PCB-180	23.52	1.65	18.09	1.34	5.93	0.24	27.52	3.03	29.47	1.47
ΣPCBs on PS (ng)		0.063		0.151		0.151		0.287		0.363	

**Table 2 toxics-08-00059-t002:** Spatial conformation of marker PCBs.

	Name	Spatial Conformation [24]	Descriptors [22] *	log Kow [25]
PCB-28	2,4,4′-Trichlorobiphenylortho, para, para	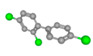	CP1, PP	5.67
PCB-52	2,2′,5,5′-Tetrachlorobiphenylortho, ortho, meta, meta	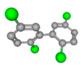	4CL, 2M	5.84
PCB-101	2,2′,4,5,5′-Pentachlorobiphenylortho, ortho, para, meta, meta	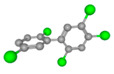	4CL, 2M	6.38
PCB-118	2,3′,4,4′,5-Pentachlorobiphenylortho, meta, para, para, meta	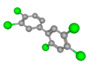	CP1, 4CL, PP, 2M	6.74
PCB-138	2,2′,3,4,4′,5′-Hexachlorobiphenylortho, ortho, meta, para, para, meta	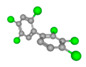	4CL, PP, 2M	6.83
PCB-153	2,2′,4,4′,5,5′-Hexachlorobiphenylortho, ortho, para, para, meta, meta	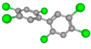	4CL, PP, 2M	6.92
PCB-180	2,2′,3,4,4′,5,5′-Heptachlorobiphenylortho, ortho, meta, para, para, meta, meta	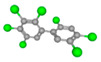	4CL, PP, 2M	7.36

Descriptors [22] *: CP0: no chlorine substitution at any of the “ortho” positions on the biphenyl backbone (also referred as non-“ortho” congeners). CP1: chlorine substitution at only one of the “ortho” positions (also referred as mono-“ortho” congeners). 4Cl: congeners that have a total of four or more chlorine substituents regardless of position. PP: congeners that have both “para” positions chlorinated. 2M: congeners that have two or more of the “meta” positions chlorinated. If one congener has all four descriptors this is referred to as being “dioxin-like”, like PCB-118.

**Table 3 toxics-08-00059-t003:** Freundlich equation parameters for ƩPCBs on MPLs in the seawater/sediment system.

	*log* [*q*] = *log K_F_* + 1/*n log* [*c*]	*log K_F_*	*K_F_* (ng/mg)	*n*
PS	y = 1.1599x + 2.4499; R^2^ = 0.9794	2.4499	281.77	0.862
PE	y = 1.1452x + 2.4258; R^2^ = 0.9253	2.4258	266.56	0.873
PET	y = 0.9532x + 2.8781; R^2^ = 0.9572	2.8781	755.27	1.049

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
