# Peer review of "Adsorption and Desorption Behaviour of Polychlorinated Biphenyls onto Microplastics’ Surfaces in Water/Sediment Systems"

_toxics, 2020, doi:10.3390/toxics8030059_

Round 1

Reviewer 1 Report

This manuscript addresses an important subject associated with microplastics and PCB contamination in a controlled water/sediment condition. Overall, the manuscript is well-written with very publishable data. Proper interpretation narrative on the collected data was made along with citation of comparative papers. I made detailed comments below, but I like to strongly suggest adding a schematics or diagram that explains PCB partitioning in the studied water/sediment systems. Similar diagram (e.g., flow chart) can be presented for experimental procedures for other scientists who like to do similar experiment. Nice work and I enjoyed reading this manuscript. Hope my comments help.

L43 Remove “nowadays”.

L46 Suggest adding general description of polymeric matrices of MPLs to educate readers.  

L50 “.. MPLs could act as a back door to the entrance…” -> Suggest revision. MPLs as a substrate to adsorb contaminants and initiate bioaccumulation on marine food web?

L58 I would spell out E.g., -> For example, it has been…..

L105 Specify how the six sediment subsamples were spiked with PCB? 1 gram of sediment was pipetted with 0, 5, 10, 15, and 25 ng of PCB mixtures? I would specify actual gram weighed for sediment and PCB, which would help those who like to replicate this type of lab procedures.

L111 I suggest adding a schematic (flowchart) and some photos that show overall experimental sequences. It is hard to follow and this would help readers understand the seawater/sediment systems treated with different levels of PCBs and types of MPLs.  

L150 Add unit for [A]t and [A]0 in Eq 1. Also, I suggest putting this equation and its narrative in M&M. From Eq 1, is there any chance for PCB to remain in solution? It looks like the equation considers PCB remained in sediment with known amount of PCB added. The remaining portion was assume to be adsorbed to microplastic.

L160 Somewhere in results and discussion, I suggest adding general explanation of PCB partitioning between sediment, water, and MP in a visual format (schematics or diagram).

L163 “As it can be seen, generally the congeners with more chlorine atoms in their molecules showed a minor tend to be adsorbed onto the PS-MPLs surfaces” -> Revise this sentence. The congeners with more chlorines -> less adsorption to PS-MPLs surfaces?

L168 “which difficult their stabilisation on plastic pore sites” -> Revise. which inhibit…?

L168-178 Elaborate more. I was lost in this narrative part.

L195 It looks like Figure 1 and Figure 2 are the same information, but in different format?

L256 What is BET? It needs to spelled out if it is the first mentioning here.

L263 Does C represent PCB concentration in which phase (water, sediment, or MP)? It is not clear if this study extracted PCB from MPs or sediments, or both. I would clarify it in M&M. I don’t want to create more work for authors but plotting PCB adsorption data onto sediment would be helpful (can be added in supporting material) as this study covered competition between sediment and MP in terms of PCB adsorption.

L294 KF -> Need subscript for F and make the same throughout the manuscript as defined by Eq 2-3.  

L280 nF -> Did authors define nF somewhere? Clarify what this is.

L292 co-planarity -> Can you explain more for readers in an easy language?

L296 Same for “steric congestion”

L311 Readers including myself would wonder how much PCB was adsorbed to plastic vs. sediment with supporting numbers.

Author Response

Reviewer 1

Open Review

English language and style

( ) Extensive editing of English language and style required 
( ) Moderate English changes required 
(x) English language and style are fine/minor spell check required 
( ) I don't feel qualified to judge about the English language and style 

Yes

Can be improved

Must be improved

Not applicable

Does the introduction provide sufficient background and include all relevant references?

(x)

( )

( )

( )

Is the research design appropriate?

(x)

( )

( )

( )

Are the methods adequately described?

( )

(x)

( )

( )

Are the results clearly presented?

( )

(x)

( )

( )

Are the conclusions supported by the results?

( )

(x)

( )

( )

Comments and Suggestions for Authors

This manuscript addresses an important subject associated with microplastics and PCB contamination in a controlled water/sediment condition. Overall, the manuscript is well-written with very publishable data. Proper interpretation narrative on the collected data was made along with citation of comparative papers. I made detailed comments below, but I like to strongly suggest adding a schematics or diagram that explains PCB partitioning in the studied water/sediment systems. Similar diagram (e.g., flow chart) can be presented for experimental procedures for other scientists who like to do similar experiment. Nice work and I enjoyed reading this manuscript. Hope my comments help.

Thanks for the revision and all the comments. We have addressed all the comments in the coming lines, including the addition of the diagram of the partitioning between phases and the flowchart of the experiment.

L43 Remove “nowadays”.

It has been removed (line 42)

L46 Suggest adding general description of polymeric matrices of MPLs to educate readers.  

It has been included, now in line 47, the next clarification: “(e.g. polyethylene, polystyrene, a combination of different monomers, among others),”

L50 “.. MPLs could act as a back door to the entrance…” -> Suggest revision. MPLs as a substrate to adsorb contaminants and initiate bioaccumulation on marine food web?

The next sentence has been added in line 52 “once they are adsorbed in the polymer surface”

L58 I would spell out E.g., -> For example, it has been…..

It has been changed accordingly (line 59)

L105 Specify how the six sediment subsamples were spiked with PCB? 1 gram of sediment was pipetted with 0, 5, 10, 15, and 25 ng of PCB mixtures? I would specify actual gram weighed for sediment and PCB, which would help those who like to replicate this type of lab procedures.

The sentence has been clarified between lines 105 - 107 according to: “Hereafter, six sediment subsamples of 4.5 g were separated, and they were spiked to the following concentrations of the PCBs mixture (0, 5, 10, 15, and 25 ng/g d.w spiked to the appropriate volume of a mixture of 10 µg/ml in methanol).”

L111 I suggest adding a schematic (flowchart) and some photos that show overall experimental sequences. It is hard to follow and this would help readers understand the seawater/sediment systems treated with different levels of PCBs and types of MPLs.

Thanks for the suggestion, we have added a schematic flowchart in the supplementary material (Figure S1) in line 113.

L150 Add unit for [A]t and [A]0 in Eq 1. Also, I suggest putting this equation and its narrative in M&M. From Eq 1, is there any chance for PCB to remain in solution? It looks like the equation considers PCB remained in sediment with known amount of PCB added. The remaining portion was assumed to be adsorbed to microplastic.

The equation has been modified accordingly and the concentration of A clarified in line 157 with the sentence “…, both concentrations expressed in ng/g.”

Regarding the adsorption of PCBs, it is assumed that they are in sediments or in MPLs because of their high logKow. According to this, it is not plausible to find them in water phase.

L160 Somewhere in results and discussion, I suggest adding general explanation of PCB partitioning between sediment, water, and MP in a visual format (schematics or diagram).

A scheme has been added as Figure S2 of the supporting material (line 159).

L163 “As it can be seen, generally the congeners with more chlorine atoms in their molecules showed a minor tend to be adsorbed onto the PS-MPLs surfaces” -> Revise this sentence. The congeners with more chlorines -> less adsorption to PS-MPLs surfaces?

It has been corrected in line 167.

L168 “which difficult their stabilisation on plastic pore sites” -> Revise. which inhibit…?

It has been clarified in line 171.

L168-178 Elaborate more. I was lost in this narrative part.

It has been changed in lines 177 - 181

L195 It looks like Figure 1 and Figure 2 are the same information, but in different format?

Yes, we have removed Figure 2 since they are repetitive.

L256 What is BET? It needs to spelled out if it is the first mentioning here.

It has been added in line 252 “…BET (Brunauer–Emmett–Teller)”

L263 Does C represent PCB concentration in which phase (water, sediment, or MP)? It is not clear if this study extracted PCB from MPs or sediments, or both. I would clarify it in M&M. I don’t want to create more work for authors but plotting PCB adsorption data onto sediment would be helpful (can be added in supporting material) as this study covered competition between sediment and MP in terms of PCB adsorption.

The concentration C is in sediment. It has been clarified now in line 259. In addition, figure S3 from the supplementary information shows the concentration of PCBs in MPLs vs the concentration in sediments after 21 days of exposition.

L294 KF -> Need subscript for F and make the same throughout the manuscript as defined by Eq 2-3.  

It has been corrected through the manuscript.

L280 nF -> Did authors define nF somewhere? Clarify what this is.

This term is for “n” in the equation 2. It has been clarified thorough the text.

L292 co-planarity -> Can you explain more for readers in an easy language?

It has been explained in line 287 as: “….co-planarity (all chlorines are in the same plane) makes…”

L296 Same for “steric congestion”

It has been explained in line 289 as: “…the steric congestion (presence of surrounding ligands (chlorines in PCBs) in a molecule)”

L311 Readers including myself would wonder how much PCB was adsorbed to plastic vs. sediment with supporting numbers.

The adsorption of sediment vs MPLs is shown in Figure S3 for each plastic type. However, if the

Reviewer 2 Report

The article is very interesting and it is focus on the current topic. I have only few minor comments and questions.

Comments:

Authors should check errors and/or typos in the whole article. There are mentioned those that I noticed : line 153 - no comma at the end of the line; line 206 - different type of chapter name as in previous case (unite it); line 239 - I am really not sure what is chapter name and what is a normal text; line 246 - the mentioned paper from Pascall et al. is without a reference number [21]; line 248 - "In the present studY..." is without Y at the end.

In the equations description (lines 154, 155, 263, 264) symbols should be in italics and you should write the lower index for marks in the text as it is in the equation.

Figure 2:

Missing x-axis name, add it to all three graphs.

Figure 3:

Axis ranges should by the same for all three graphs A, B, and C for easier mutual comparison. Adjust the scales on the y-axis for all three graphs; numbers with four zero - it is not a scientific label. Images size (optically) is different (Fig. 3A > Fig. 3C), unite it.

The names of each graph are useless. You marked individual graphs by symbols A, B, and C and in the description of Figure you explained it. (the same goes for Figure 2).

Questions:

Figure 2A: Can you explain a sharp decline of adsorption all PCBs on PS at the concentration 15ng/g.  It is different behavior as for PE and PET, moreover, at the higher concentration (>15ng/g) the adsorption of PCBs on PS again rapidly increase. Why?

Line 221-223: “For example for the PCB 118, the maximum adsorption level was 10 % for the spiking of 15 ng/g…” This statement does not match with Figure 2B, where the maximum adsorption level was about 15 % for the concentration 15 ng/g.  Do you have an error in the graph or in the text or somewhere else?

Figure 3A: How do you explain the more significant differences between the experimental points and theoretical fitted Freundlich model for adsorption isotherms on PS (moreover, when this difference is not so obvious for PE and PET).

How many times were repeated the experimental measurements? Are the presented results the average of several measurements or it is only absolute numbers from one series of experiment? In both cases, it would be good adding this information to the article.

Author Response

Reviewer 2

Open Review

English language and style

( ) Extensive editing of English language and style required 
( ) Moderate English changes required 
( ) English language and style are fine/minor spell check required 
(x) I don't feel qualified to judge about the English language and style 

Yes

Can be improved

Must be improved

Not applicable

Does the introduction provide sufficient background and include all relevant references?

(x)

( )

( )

( )

Is the research design appropriate?

(x)

( )

( )

( )

Are the methods adequately described?

(x)

( )

( )

( )

Are the results clearly presented?

(x)

( )

( )

( )

Are the conclusions supported by the results?

(x)

( )

( )

( )

Comments and Suggestions for Authors

The article is very interesting and it is focus on the current topic. I have only few minor comments and questions.

Thanks for the revision. We have addressed all the comments in the next section.

Comments:

Authors should check errors and/or typos in the whole article. There are mentioned those that I noticed : line 153 - no comma at the end of the line; line 206 - different type of chapter name as in previous case (unite it); line 239 - I am really not sure what is chapter name and what is a normal text; line 246 - the mentioned paper from Pascall et al. is without a reference number [21]; line 248 - "In the present studY..." is without Y at the end.

Thanks for the comments, we have corrected all of them through the text

In the equations description (lines 154, 155, 263, 264) symbols should be in italics and you should write the lower index for marks in the text as it is in the equation.

It has been corrected accordingly thorough the text

Figure 2:

Missing x-axis name, add it to all three graphs.

This figure has been deleted according to Reviewer 1 comments

Figure 3:

Axis ranges should by the same for all three graphs A, B, and C for easier mutual comparison. Adjust the scales on the y-axis for all three graphs; numbers with four zero - it is not a scientific label. Images size (optically) is different (Fig. 3A > Fig. 3C), unite it.

The names of each graph are useless. You marked individual graphs by symbols A, B, and C and in the description of Figure you explained it. (the same goes for Figure 2).

All the graphs have been corrected according to reviewer comments

 Questions:

Figure 2A: Can you explain a sharp decline of adsorption all PCBs on PS at the concentration 15ng/g.  It is different behavior as for PE and PET, moreover, at the higher concentration (>15ng/g) the adsorption of PCBs on PS again rapidly increase. Why?

Although we have removed Figure 2, the same behavior can be seen in Figure 1. Our explanation is because of experimental error where maybe the diffusion of PS in sediments has not been so efficient as in the other cases. Nonetheless, this graph reflects the results of the media of three analyses.

Line 221-223: “For example for the PCB 118, the maximum adsorption level was 10 % for the spiking of 15 ng/g…” This statement does not match with Figure 2B, where the maximum adsorption level was about 15 % for the concentration 15 ng/g.  Do you have an error in the graph or in the text or somewhere else?

Thanks for the comment, this was an error in the text. The correct percentage is 15%. Now this is corrected in line 217.

Figure 3A: How do you explain the more significant differences between the experimental points and theoretical fitted Freundlich model for adsorption isotherms on PS (moreover, when this difference is not so obvious for PE and PET).

The main hypotheses are the experimental error in the experiments with 15 ng/g or the bad diffusion between PS and sediments.

How many times were repeated the experimental measurements? Are the presented results the average of several measurements or it is only absolute numbers from one series of experiment? In both cases, it would be good adding this information to the article.

The experiments were done in triplicate. This is now added in the text, line 134
